Understanding fatality patterns and sex ratios of Brazilian free-tailed bats (Tadarida brasiliensis) at wind energy facilities in western California and Texas

LiCari Sarah T. 1
http://orcid.org/0000-0001-9701-0763 Hale Amanda M. 1 2
http://orcid.org/0000-0001-5555-5087 Weaver Sara P. 3
Fritts Sarah 4
Katzner Todd 5
http://orcid.org/0000-0003-2755-5535 Nelson David M. 6
http://orcid.org/0000-0002-9001-6019 Williams Dean A. 1 dean.williams@tcu.edu
1 Department of Biology, Texas Christian University , Fort Worth, Texas , United States
2 Western EcoSystems Technology, Inc , Cheyenne, Wyoming , United States
3 Bowman Consulting Group , San Marcos, Texas , United States
4 Department of Biology, Texas State University , San Marcos, Texas , United States
5 U.S. Geological Survey, Forest and Rangeland Ecosystem Science Center , Boise, Idaho , United States
6 University of Maryland Center for Environmental Science, Appalachian Laboratory , Frostburg, Maryland , United States
Mahmood Haider
Electronic publication date: 2023 Dec 7
Publication date: 2023
Volume: 11
Electronic Location ID: e16580
Received 2023 Aug 25; Accepted 2023 Nov 13
Copyright: © 2023 LiCari et al.
Copyright year: 2023
Copyright holder: LiCari et al.
License: This is an open access article distributed under the terms of the Creative Commons Attribution License, which permits unrestricted use, distribution, reproduction and adaptation in any medium and for any purpose provided that it is properly attributed. For attribution, the original author(s), title, publication source (PeerJ) and either DOI or URL of the article must be cited.
License URL: https://creativecommons.org/licenses/by/4.0/

Keywords: Brazilian free-tailed bat, Tadarida brasiliensis, Wind energy, Wind turbines, Fatality monitoring, Sex ratios, California, Texas

Funding: Renewable Energy Wildlife Research Fund B-19 Texas Chrisitan University Adkin’s grant 2022-01 This work was supported by the Renewable Energy Wildlife Research Fund (No. B-19) and the Texas Chrisitan University Adkin’s grant (No. 2022-01). The funders had no role in study design, data collection and analysis, decision to publish, or preparation of the manuscript.

==============================
Background

Operation of wind turbines has resulted in collision fatalities for several bat species, and one proven method to reduce these fatalities is to limit wind turbine blade rotation (i.e., curtail turbines) when fatalities are expected to be highest. Implementation of curtailment can potentially be optimized by targeting times when females are most at risk, as the proportion of females limits the growth and stability of many bat populations. The Brazilian free-tailed bat (Tadarida brasiliensis) is the most common bat fatality at wind energy facilities in California and Texas, and yet there are few available data on the sex ratios of the carcasses that are found. Understanding the sex ratios of fatalities in California and Texas could aid in planning population conservation strategies such as informed curtailment.

Methods

We used PCR to determine the sex of bat carcasses collected from wind energy facilities during post-construction monitoring (PCM) studies in California and Texas. In California, we received samples from two locations within the Altamont Pass Wind Resource Area in Alameda County: Golden Hills (GH) (n = 212) and Golden Hills North (GHN) (n = 312). In Texas, we received samples from three wind energy facilities: Los Mirasoles (LM) (Hidalgo County and Starr County) (n = 252), Los Vientos (LV) (Starr County) (n = 568), and Wind Farm A (WFA) (San Patricio County and Bee County) (n = 393).

Results

In California, the sex ratios of fatalities did not differ from 50:50, and the sex ratio remained stable over the survey years, but the seasonal timing of peak fatalities was inconsistent. In 2017 and 2018, fatalities peaked between September and October, whereas in 2019 and 2020 fatalities peaked between May and June. In Texas, sex ratios of fatalities varied between locations, with Los Vientos being female-skewed and Wind Farm A being male-skewed. The sex ratio of fatalities was also inconsistent over time. Lastly, for each location in Texas with multiple years studied, we observed a decrease in the proportion of female fatalities over time.

Discussion

We observed unexpected variation in the seasonal timing of peak fatalities in California and differences in the sex ratio of fatalities across time and facility location in Texas. In Texas, proximity to different roost types (bridge or cave) likely influenced the sex ratio of fatalities at wind energy facilities. Due to the inconsistencies in the timing of peak female fatalities, we were unable to determine an optimum curtailment period; however, there may be location-specific trends that warrant future investigation. More research should be done over the entirety of the bat active season to better understand these trends in Texas. In addition, standardization of PCM studies could assist future research efforts, enhance current monitoring efforts, and facilitate research on post-construction monitoring studies.

Introduction

Portions of this text were previously published as part of a thesis (LiCari, 2023). Renewable energy production has expanded in recent decades to increase energy production and decrease dependence on fossil fuels, which are a significant source of carbon emissions (Saidur et al., 2011; Barthelmie & Pryor, 2014; International Energy Agency (IEA), 2022). Wind energy is one of the fastest-growing renewable energy industries in the world (International Energy Agency (IEA), 2022). While there are many recognized benefits of wind energy (Wiser et al., 2015), an unintended consequence has been the negative impacts on birds and bats (Kuvlesky et al., 2007; Arnett & May, 2016; Allison et al., 2019; American Wind Wildlife Institute (AWWI), 2020). These impacts include displacement, habitat fragmentation, and collision mortality (Drewitt & Langston, 2006; Arnett & May, 2016; Allison et al., 2019; Diffendorfer et al., 2019).

For bat fatalities in North America, migratory tree-roosting species such as the hoary (Aeorestes cinereus), silver-haired (Lasionycteris noctivagans), and eastern red (Lasiurus borealis) bat comprise more than 75% of known fatalities (American Wind Wildlife Institute (AWWI), 2020; Western EcoSystems Technology, Inc. (WEST), 2023). Estimates indicate that annual fatalities are in the hundreds of thousands in the United States (US) and Canada (Arnett & Baerwald, 2013; Zimmerling & Francis, 2016; American Wind Wildlife Institute (AWWI), 2020). As bats are long-lived and have slow reproductive rates, they are especially susceptible to population declines (Barclay & Fleming, 2020). Collision mortality could lead to severe population declines or even extinction for some species, depending on starting population sizes and important demographic parameters which are largely unknown (Frick et al., 2017; Friedenberg & Frick, 2021).

Some biologists have also expressed concern for Brazilian free-tailed bats (Tadarida brasiliensis) with respect to wind energy development (Weaver et al., 2020a). This species is the fourth most common bat species found in post-construction fatality monitoring (PCM) studies at wind energy facilities (American Wind Wildlife Institute (AWWI), 2020; Western EcoSystems Technology, Inc. (WEST), 2023), with most fatalities occurring in the late summer and early autumn (Arnett et al., 2008; Allison et al., 2019; American Wind Wildlife Institute (AWWI), 2020). By US Fish and Wildlife Region, the Brazilian free-tailed bat is the most common bat species found during PCM studies in the Southwest and Pacific Southwest of the US, at 39% and 66%, respectively (Western EcoSystems Technology, Inc. (WEST), 2023). However, it should be noted that the overall number of fatalities in these regions is lower than in other regions (American Wind Wildlife Institute (AWWI), 2020). The Brazilian free-tailed bat is a partially migratory cave bat that lives in colonies, with some numbering up to in the millions (Glass, 1958; Villa & Cockrum, 1962; Davis, Herreid & Short, 1962; Hristov et al., 2010). These large populations are a source of agricultural pest control (Federico et al., 2008; Boyles et al., 2011) and ecotourism (Bagstad & Wiederholt, 2013). For example, the Brazilian free-tailed bat is estimated to reduce cotton crop damage by up to 43% in south-central Texas (Federico et al., 2008), and their large colonies contribute >$6.5 million in consumer surplus across six states in the Southwestern US on an annual basis (Bagstad & Wiederholt, 2013).

Given the importance of the Brazilian free-tailed bat, there is interest in conserving their populations which have already experienced declines due to pesticide use (Geluso, Altenbach & Wilson, 1976, 1981; Clark, 2001) and human disturbances to maternal roosts (Furey & Racey, 2016) at locations such as Carlsbad Cavern in New Mexico (Clark, 2001) and Eagle Creek Cave in Arizona (Cockrum, 1970). To understand the potential impacts of wind turbine collision mortality on Brazilian free-tailed bat populations, knowledge of the sex ratio of fatalities could be used to inform conservation strategies for this partially migratory cave species. A proven bat fatality reduction strategy is curtailing wind turbine operations during periods of risk (reviewed in Adams, Gulka & Williams, 2021; Whitby, Schirmacher & Frick, 2021). This method has been further refined by incorporating additional predictor variables to further define periods of risk, such as temperature (Martin et al., 2017), and near real-time curtailment decisions based on the presence of bat activity (Hayes et al., 2019). Data on the sex ratio of fatalities could also be used for informed curtailment. With data on the proportion of female fatalities during the bat active season, we could emphasize curtailment during periods of peak female fatalities since females are the limiting factor for population growth and stability in bats (Grüebler et al., 2008; Wedekind, 2012). Furthermore, if female-focused curtailment periods are only a subset of the peak fatality season, this could incentivize wind facility operators to implement curtailment measures for the Brazilian free-tailed bat if the expected power loss is less than it otherwise would be under longer periods of curtailment.

In this study, we determined the proportion of female fatalities of Brazilian free-tailed bats at wind energy facilities during year-long PCM studies across multiple years in California and Texas, where the Brazilian free-tailed bat is the most common fatality at wind energy facilities (American Wind Wildlife Institute (AWWI), 2020; Western EcoSystems Technology, Inc. (WEST), 2023). Populations from these two states were also of interest due to differences in migratory behavior and population sex ratios (Villa & Cockrum, 1962; Davis, Herreid & Short, 1962; Glass, 1982; Ammerman, Hice & Schmidly, 2012; Wiederholt et al., 2013). The western California population is non-migratory and hibernates during the winter; and the sexes do not exhibit sex-specific movement across the landscape (Krutzsch, 1955; Davis, Herreid & Short, 1962). For this reason, we do not expect the proportion of female fatalities to differ across seasons. From this point forward, we will refer to this population as either the California population or the non-migratory population. Although the sex ratio of the population is unknown, it may be assumed to be 50:50 (F:M), as seen in other non-migratory bat species populations (Davis, 1969; Cheng et al., 2019). In this assumed 50:50 population, the sex ratio of fatalities could be influenced by sex-specific behaviors around wind turbines, such as sex-specific attraction to turbines (Cryan, 2008; Arnett et al., 2008; Guest et al., 2022) or sex-specific foraging strategies (Safi, König & Kerth, 2007; Istvanko, Risch & Rolland, 2016). If the proportion of female fatalities differs from 0.5, then this could indicate that female and male bats in the Californian population interact differently with wind turbines. In Texas, the Brazilian free-tailed bat population migrates annually between Mexico and the US to form large maternity colonies in the US during spring and summer, with the migrant population being female-skewed (Villa & Cockrum, 1962; Davis, Herreid & Short, 1962; Glass, 1982; McCracken, 2003; Lopéz-González, Rascón & Daniel Hernández-Velázquez, 2010; Ammerman, Hice & Schmidly, 2012; Wiederholt et al., 2013). Davis, Herreid & Short (1962) sexed more than 40,000 Brazilian free-tailed bats in Texas, revealing a substantial female bias in the migratory population with a ratio of 9:1 (Female:Male). While information is available about the life histories of the California and Texas populations, limited knowledge exists regarding the impact of wind energy on these populations, particularly regarding the sex ratios of fatalities.

In summary, our goal was to determine the sex ratios of Brazilian free-tailed bat fatalities at wind energy facilities in California and Texas and to determine if the sex ratio of fatalities was stable over time. We expected the timing of peak fatalities to occur between September and October in California and Texas due to the addition of pups to the flying populations. In Texas, this time frame also overlaps with the return migration of the Brazilian free-tailed bat to Mexico to overwinter (Villa & Cockrum, 1962; Davis, Herreid & Short, 1962; Glass, 1982; Ammerman, Hice & Schmidly, 2012; Wiederholt et al., 2013). We also expected to see differences in the proportion of female fatalities between California and Texas. Given the differences in the proportion of females and migratory patterns between the two populations of Brazilian free-tailed bats, we predicted that: (1) the proportion of female fatalities in the California population would not differ from 0.5, and (2) the proportion of female fatalities in the Texas population would be >0.5.

Materials and Methods

Carcass collection

For all tissue samples, we requested information on the collection date, location, estimated time since death, species, and sex (male, female, unknown). Species and sex were determined in the field using morphological characteristics. While sex was provided for some bat samples based on morphological characteristics, all bats were sexed using molecular methods for our analysis. Molecular methods allowed us to obtain more data by sexing samples that could not be identified in the field and provided more accurate understanding of the sex ratios of fatalities (Chipps et al., 2020).

California

Bat carcasses were collected from two wind energy facilities (Golden Hills and Golden Hills North) by H.T. Harvey & Associates (HTH) under a permit from the California Department of Fish and Wildlife (SC-4607). Carcasses were sent to the Forest and Rangeland Ecosystem Science Center under the California Scientific Collection Permit SC-11910 and the Idaho Wildlife Collection/Banding/Possession Permit 110728. Texas Christian University received wing tissue samples from these carcasses under the Texas Parks and Wildlife Department (TPWD) permit number SPR-1211-390.

Golden Hills (GH) comprises 48 1.79-megawatt (MW) General Electric (GE) turbines with a hub height of 80 m, a rotor diameter of 100 m, and a rotor sweep zone of 30–130 m above ground level. Habitat throughout the facility is characterized by rolling hills, primarily with grazed grasslands and sparsely scattered trees and shrubs in intervening drainages (H. T. Harvey & Associates (HTH), 2021). Climate is moderate, with warm, dry summers and cool, wet winters (Kauffman, 2003). Post-construction monitoring at GH covered three search years (Year 1: 19 September 2016–17 September 2017; Year 2: 18 September 2017–16 September 2018; Year 3: 17 September 2018–15 September 2019). A generalized random-tessellation stratified (GRTS) sampling algorithm (Stevens & Olsen, 2004) was used to ensure balanced annual sampling and spatially and temporally representative 7-day interval sampling of all turbines throughout the study. Generalized random-tessellation stratified sampling divided the study area into non-overlapping cells, stratified these cells based on specific characteristics, and then selected sample points systematically within each stratum with randomization to reduce bias. This approach ensures that the resulting sample is statistically robust and spatially representative. Each year, one-third of the turbines were searched at a seven-day interval, and the remaining two-thirds were searched at 28-day intervals. Detection dog teams searched the 7-day interval plots in all three years. For the 28-day interval plots, detection dog teams were used in Year 1 (H. T. Harvey & Associates (HTH), 2021). Human technicians searched the 28-day interval plots in Year 2 and Year 3. Detection-dog surveys were conducted considering wind patterns, with the handler using various cues, such as body movements, hand signals, verbal commands, and whistles, to guide the dog’s search in relation to wind speed and direction, ensuring complete coverage of designated survey areas. The search approach was adaptable, allowing the handler to adjust the dog’s actions as needed. The number of survey passes and detection distances were influenced by wind speed, with more passes at closer intervals in light wind and fewer passes spaced further apart in strong wind conditions. Wing tissue samples were stored in NaCl/DMSO preservative solution (6 M NaCl and 20% DMSO) in 2 ml screw-cap plastic tubes.

Golden Hills North (GHN) comprises 20 2.3-MW GE turbines, with a hub height of 80 m, a rotor diameter of 110 m, and a rotor sweep zone of 25–135 m above ground level. Habitat throughout the facility is characterized by rolling hills with occasional steep slopes and rocky outcrops, grazed grasslands, small groves, and seasonal livestock ponds (Great Basin Bird Observatory (GBBO) & H. T. Harvey & Associates (HTH), 2022). Climate is moderate, with warm, dry summers and cool, wet winters (Kauffman, 2003). Post-construction monitoring at GHN covered three search years from 8 October 2018–29 September 2021. Similar to GH, generalized random-tessellation stratified (GRTS) sampling was used (Stevens & Olsen, 2004). Each year, half of the turbines were searched at a seven-day interval, and the other half were searched at 28-day intervals. Detection dog teams searched the seven-day and 28-day interval plots across all years (Great Basin Bird Observatory (GBBO) & H. T. Harvey & Associates (HTH), 2022). Detection dog surveys followed similar procedures as those mentioned above. Wing tissue samples were stored in NaCl/DMSO preservative solution (6 M NaCl and 20% DMSO) in 2 ml screw-cap plastic tubes.

Texas

Bat carcasses were collected by Texas State University or Bowman Consulting Group in accordance with Texas Parks & Wildlife Department (TPWD) permit numbers SPR-1120-189 and SPR-0213-023. Texas Christian University received the wing tissue samples under the TPWD permit number SPR-1211-390. Los Mirasoles (LM) comprises 125 2.0-MW Vestas V-110 turbines, with a hub height of 80 m, a rotor diameter of 110 m, and a rotor sweep zone of 30–130 m above ground level. Habitat throughout the facility is characterized as largely flat with primarily cultivated crops, followed by shrub-scrub and pasture/hay (Texas Parks & Wildlife Department (TPWD), 2015). Climate in this region is typically semi-arid. Post-construction monitoring at LM was conducted on a weekly basis during spring, summer, and fall (3 August 2017–25 November 2017 and 25 February–31 July 2018) and bi-weekly during winter (26 November 2017–24 February 2018) at 100 turbines selected through stratified random sampling and searched at a seven-day interval with human technicians. Of the 100 turbines searched, eight had full plots with a radius of 100 m, and the remaining 92 were only searched on gravel roads and pads. Wing tissue samples were stored in 95% ethanol in 2 ml screw-cap plastic tubes.

Los Vientos (LV) comprises 255 2.0 MW Vestas V-110 turbines, with a hub height of 95 m, a rotor diameter of 110 m, and a rotor sweep zone of 40–150 m above ground level. Habitat at the facility consisted of shrub-scrub, cultivated croplands, and pasture/hay (Weaver et al., 2020a). The eastern portion of the facility is largely flat, with rolling hills occurring in the western portion. The search efforts varied at this facility depending on the turbine and study objectives. For standard PCM studies, 100 turbines were selected through stratified random sampling and searched by human technicians at a seven-day interval from 24 March 2017–25 November 2017, and 25 February 2018–23 March 2018, and on a 14-day interval from 26 November 2017–24 February 2018 (Weaver et al., 2020a). Similar to LM, eight of these 100 turbines had full, 100-m radius plots, and the remaining were only searched on gravel roads and pads. For a concurrent study testing efficacy of deterrents for reducing bat fatalities that occurred from 1 August to 31 October in both 2017 and 2018, 100-m radius plots at 16 randomly selected turbines were searched daily, seven days a week, by human technicians (Weaver et al., 2020b). Wing tissue samples, approximately 1 cm2, were stored in 95% ethanol in 2 ml screw-cap plastic tubes.

Wind Farm A (WFA) comprises 124 wind turbines, of which 93 were GE 2.52-MW turbines (height: 88.6 m; rotor diameter: 127 m; rotor sweep zone: 25.1–152.1 m), 22 were GE 2.3-MW turbines (height: 90 m; rotor diameter: 116 m; rotor sweep zone: 32–148 m), and nine were GE 2.5-MW turbines (height: 90 m; rotor diameter: 116 m; rotor sweep zone: 31–148 m). Habitat at the facility is primarily flat with cultivated croplands. Gravel roads and pads were searched weekly at all turbines by human technicians from 2 November 2020–14 May 2021 and 16 July 2021–27 October 2021, and every other week from 15 May 2021–15 July 2021. Gravel roads and pads were also searched weekly from 18 July 2022–31 October 2022. Wing tissue samples were stored in NaCl/DMSO preservative solution (6 M NaCl and 20% DMSO) in 2 ml screw-cap plastic tubes.

DNA extraction

Portions of this text were previously published as part of a thesis (LiCari, 2023). We extracted DNA from the wing tissue samples using 96-well PCR plates, following the methods detailed in Ivanova, Dewaard & Herbert (2006), with an extra wash step. We placed a small piece of each sample (approx. 2–3 mm2) in an individual well in a 96-well PCR plate. Each well contained 50 μl of Vertebrate Lysis Buffer (VLB) (100 mM NaCl, 50 mM Tris-HCL pH 8.0, 10 mM EDTA pH 8.0, and 0.5% SDS) and 5 μl of Proteinase K (20 mg/ml). We then covered the 96-well PCR plate with a PCR mat and incubated it at 56 °C overnight to allow digestion. After the wing tissue was digested, we centrifuged the 96-well PCR plate at 1,500 g for 10 s to remove condensation from the PCR mat. Next, we added 100 μl of Binding Mix (BM), made from equal volumes of EtOH 96% and Binding Buffer (BB) (6 M GuSCN, 20 mM EDTA pH 8.0, 10 mM Tris-HCL pH 6.4, 4% Triton X-100) to each sample. We then covered the 96-well PCR plate with a new PCR mat, vortexed the plate vigorously for 15 s, and centrifuged the 96-well PCR plate at 1,000 g for 20 s. We transferred the lysate (about 150 μl) from the wells of the 96-well PCR plate into the wells of a Glass Fiber (GF) Plate (PALL1) placed on top of a square-well block. We sealed the GF Plate with a self-adhering cover. The covered GF Plate and square-well block were centrifuged at 5,000 g for 5 min to bind the DNA to the GF membrane. Three wash steps followed. For the first wash step, we added 180 μl of Protein Wash Buffer (PWB) (46.8 μl BB, 126 μl EtOH 96%, and 7.2 μl autoclaved dH2O) to each well of the GF Plate, sealed the GF Plate with a new self-adhering cover, and centrifuged the GF Plate and square-well block at 5,000 g for 2 min. For the second wash step, we added 750 μl of Wash Buffer (WB) (60% EtOH 96%, 50 mM NaCl, 10 mM Tris-HCL pH 7.4, and 0.5 mM EDTA pH 8.0) to each well of the GF Plate, sealed the plate with a new self-adhering cover, and centrifuged the GF Plate and square-well block at 5,000 g for 5 min. For the third wash step, we added 500 μl of 70% ethanol to each well of the GF Plate, sealed the plate with a new self-adhering cover, and centrifuged the GF Plate and square-well block at 5,000 g for 3 min. After the wash steps, we centrifuged the GF Plate and square-well block at 5,000 g for 5 min to remove excess moisture from the samples. We then placed the GF Plate on top of a microplate equipped with a PALL collar and dispensed 70 μl of 10 mM Tris-HCL, pH 8.5 (pre-warmed to 56 °C) directly onto the membrane in each well of the GF Plate. We then incubated the GF Plate on top of a microplate equipped with a PALL collar at room temperature for 1 min. After incubating at room temperature, we covered the GF Plate with a new self-adhering cover. We then centrifuged the assembled plate at 5,000 g for 5 min to collect the DNA eluate into the bottom microplate. We then sealed the DNA plate with a new self-adhering cover and stored the extracted samples at −20 °C.

Molecular sex determination

We determined sex by amplifying the zfx and zfy introns found on the sex chromosomes using the Brazilian free-tailed primers in Korstian et al. (2013). All PCRs (10 μl) contained between 2 to 50 ng/μl of template DNA, 0.5 μM of each X-primer, 0.35 μM of each Y-primer, and 1X AccuStart II™ PCR SuperMix. The cycling parameters for the PCRs were one cycle at 95 °C for 15 min, followed by 35 cycles of 30 s at 94 °C, 1 min at 57 °C, and 30 s at 72 °C. We stained PCR products using Gel Green (Biotium), electrophoresed at 200 V in 1% agarose gel for 35 min and visualized the bands using blue-green LED light.

We used the number of bands present to determine sex. Females produce a single band corresponding to the X-chromosome intron (245 bp). Males produce two bands, one corresponding to the X-chromosome intron (245 bp) and one corresponding to the Y-chromosome intron (80 bp). We disregarded bands larger than ~300 bp when determining sex.

Data analysis

The groupings we analyzed were location (GH; GHN; LM; LV; WFA), year by location, and month by location. We were interested in determining whether the proportion of female fatalities differed from 0.5 for these groupings and whether the proportion of female fatalities changed over time (year and month) by location. Due to variation in search efforts across locations and time, aspects of data analysis differed between California and Texas, and different constraints were applied to each dataset. We will discuss these differences and provide more detailed information on the statistical tests used in subsequent paragraphs.

To compare sex ratios, we used carcass count data rather than fatality estimates, such as those obtained from GenEst, that take into account searcher efficiency, carcass persistence times, and density-weighted proportion (DWP) of the area searched (Dalthorp et al., 2018a, 2018b). Estimators like GenEst adjust the observed carcass counts to generate more accurate estimates of fatality that account for these biases. Nonetheless, we did not expect these biases to impact estimated sex ratios based on carcass counts. Previous studies suggested that carcass removal by scavengers at wind energy facilities was not sex-specific, so we expected each sex to remain available to be detected for equal amounts of time (Villegas-Patraca et al., 2012; DeVault et al., 2017). Similarly, we also did not expect searcher efficiency to influence the sex ratios. The Brazilian free-tailed bat is not sexually dimorphic (Twente, 1956; Davis, Herreid & Short, 1962; Kunz & Robson, 1995), so we expected each sex to be equally detectable to both detection dog teams and technicians. Searcher efficiency could change between seasons due to a change in environmental conditions that influenced detectability. This could result in sex ratios that are biased towards seasons when searches were less difficult, and more carcasses were collected. For example, the extreme heat of Texas summers could have decreased searcher efficiency during this season, which in turn could have led to an underrepresentation of data from the summer months when calculating annual sex ratios. This bias could be an issue for the Texas population, where we expected a difference in the sex ratio of fatalities across seasons. However, in Texas, we only analyzed data from August to October, so we did not analyze all seasons. Lastly, we did not expect the DWP to bias the sex ratios. Since the sexes are similar in mass (Twente, 1956; Davis, Herreid & Short, 1962; Kunz & Robson, 1995), we would not expect the carcass fall distributions to differ between the sexes.

For California, the groups used for analysis were year by location (GH, GHN) and month by location (GH, GHN). For the by-year analysis, we analyzed data only for years, with searches conducted from January–December. With this restriction, the two locations had no data overlap for the by-year analysis, excluding GH 2017, GH 2019, and GHN 2018 due to partial search years. We separated the month analysis by location and only analyzed months with combined totals of ≥30 bats pooled across years at the location (GH: September, October; GHN: May, June, July, August, September, October). Thirty was chosen as the minimum sample size to align with the central limit theorem. We used a one-proportion z-test for all groupings to determine if the proportion of female fatalities differed from 0.5 for location, years by location, and months by location. We conducted pairwise comparisons using multiple two-proportion z-tests, with a Bonferroni correction when applicable, to determine if the proportion of female fatalities differed between locations (GH vs. GHN), years by location (GH year vs. year; GHN year vs. year), and months by location (GH month vs. month; GHN month vs. month). In the California data analysis, we applied the Bonferroni correction to the GHN (month vs. month) analysis because the analysis required multiple pairwise comparisons within the group.

In Texas, search effort varied across locations and years. Since most searches were conducted at facilities from August to October, our analysis focused exclusively on bat fatalities during that period. This time frame also coincided with peak bat fatalities. We did not investigate the overall proportion of female fatalities by month due to differences in survey efforts across sites and seasons. The groups analyzed were location (LM, LV, and WFA) and year by location (LM 2017; LV 2017–2018; WFA 2021–2022). We used a one-proportion z-test for all groupings to determine if the proportion of female fatalities differed from 0.5. We used pairwise comparisons using multiple two-proportion z-tests, with a Bonferroni correction when applicable, to determine if the proportion of female fatalities differed between locations (LM vs. LV vs. WFA). For locations with two years of data, we used a two-proportion z-test to determine if the sex ratio changed between years at the location. In the Texas data analysis, we applied the Bonferroni correction to the comparisons between locations because the analysis required multiple pairwise comparisons within the group.

Results

We obtained 1,737 wing tissue samples (CA: GH, n = 212; GHN = 312; TX: LM, n = 252; LV, n = 568; WFA, n = 393). Of these, we were able to determine the sex of 1,647 (95%). We were unable to determine the sex for 90 bat tissue samples due to poor DNA quality. For the samples that amplified successfully, the amplification produced a product specific to the X-chromosome (245 bp) in both males and females and another product specific to the Y-chromosome (80 bp) in just males.

California

Although month-specific sample sizes were fairly small for the presumed non-migratory California population, we did not detect a difference in the proportion of female fatalities from 0.5 for year by location (1.14 > |z| > 0.44, 0.25 < p < 0.66, α = 0.05) or month by location (1.18 > |z| > 0.00, 0.24 < p < 1.00, α = 0.05), except for June at GHN (pfemale = 0.33, n = 55, z = −2.56, p = 0.01, α = 0.05). We did not detect differences among groupings: locations (GH vs. GHN, z = −0.65, p = 0.52, α = 0.05), years by location (GH year vs. year, z = 0.19, p = 0.85, α = 0.05; GHN year vs. year, z = 0.76, p = 0.45, α = 0.05), and months by location (GH month vs. month, z = −0.31, p = 0.76, α = 0.05; GHN month vs. month, 1.66 > |z| > 0.11, 0.01 < p < 0.92, α = 0.008).

One unexpected finding from the analysis was that the California population experienced a difference in the timing of peak fatalities across the facilities surveyed. The shift in the timing of peak fatalities we observed in the raw carcass counts was also evident in the GenEst corrected seasonal fatality estimates in the PCM reports for GH (H. T. Harvey & Associates (HTH), 2021) and GHN (Great Basin Bird Observatory (GBBO) & H. T. Harvey & Associates (HTH), 2022). For the first two complete years of surveys at GH (2017 and 2018), fatalities peaked between September and October, as expected (American Wind Wildlife Institute (AWWI), 2020; Figs. 1B, 1C). In 2019 and 2020, however, fatalities at GHN peaked between May and June (Figs. 1F, 1G).

Figure 1 Monthly counts of Brazilian free-tailed bat carcasses by sex found at California wind energy facilities.

Golden Hills (GH) (2016–2019, A–D) and Golden Hills North (GHN) (2018–2020, E–G).

Texas

For the migratory Texas population, the proportion of female fatalities varied among the three wind energy facilities. The proportion of female fatalities at Los Mirasoles did not differ from 0.5 (pfemale = 0.56, n = 124, z = 1.44, p = 0.150, α = 0.05; Fig. 2A). In contrast, fatalities at Los Vientos were female-skewed (pfemale = 0.55, n = 516, z = 2.25 p = 0.024; α = 0.05; Fig. 2A); and fatalities at WFA were male-skewed (pfemale = 0.41, n = 341, z = −3.20, p = 0.001, α = 0.05; Fig. 2A).

Figure 2 The proportion of female Brazilian free-tailed bat fatalities found at Texas wind energy facilities.

(A) Comparison of the proportion of female fatalities within three Texas facilities (Los Mirasoles, Los Vientos, and Wind Farm A) to a reference proportion of 0.5 (representing a 50% female fatality rate). The proportions are presented with 95% confidence intervals, and a dashed line indicates the reference value. Overlapping confidence intervals with the dashed line indicate a lack of difference from even sex ratios. (Los Mirasoles (pfemale = 0.56, n = 124); Los Vientos (pfemale = 0.55, n = 515); Wind Farm A (pfemale = 0.41, n = 341)). (B) Comparison of the proportion of female fatalities between Texas locations (Los Mirasoles, Los Vientos, and Wind Farm A). Confidence intervals are displayed at 98.3%, adjusted for multiple pairwise comparisons. Overlapping confidence intervals with the mean of other location data points indicate a lack of difference between facilities. Figures denoting statistically significant differences (α = 0.017) are marked with asterisks. (Los Mirasoles (pfemale = 0.56, n = 124); Los Vientos (pfemale = 0.55, n = 515); Wind Farm A (pfemale = 0.41, n = 341).

When comparing the proportion of female fatalities between the three locations, the proportion of female fatalities at WFA was lower than the proportion of female fatalities at Los Mirasoles (z = −2.88, p = 0.004, α = 0.017) and Los Vientos (z = −3.90, p = 0.0001, α = 0.017; Fig. 2B).

The proportion of female fatalities varied between years at wind energy facilities in Texas (Fig. 3). At Los Vientos, fatalities were female-skewed in 2017 (pfemale = 0.60, n = 285, z = 3.38, p = 0.0007, α = 0.05) but not in 2018 (pfemale = 0.49, n = 230, z = −0.40, p = 0.69, α = 0.05). The proportion of female fatalities at Los Vientos decreased by 0.11 between 2017 and 2018 (z = −2.55, p = 0.01, α = 0.05). Lastly, the peak in Brazilian free-tailed bat fatalities consistently occurred in late summer and early autumn at the Texas wind energy facilities included in this study (Fig. 4).

Figure 3 The proportion of female Brazilian free-tailed bat fatalities in Texas by location and year.

Comparison of the proportion of female fatalities in Texas facilities over different years: Los Mirasoles (2017), Los Vientos (2017 and 2018), and Wind Farm A (2021 and 2022), with 95% confidence intervals. A dashed line represents a 0.5 proportion of female fatalities, indicating a 50% female fatality rate. Confidence intervals that overlap with the dashed line indicate a lack of difference from even sex ratios. (Los Mirasoles, 2017 (pfemale = 0.56, n = 124); Los Vientos, 2017 (pfemale = 0.60, n = 285); Los Vientos, 2018 (pfemale = 0.49, n = 230); Wind Farm A, 2021 (pfemale = 0.44, n = 102); Wind Farm A, 2022 (pfemale = 0.40, n = 239)).

Figure 4 Monthly counts of Brazilian free-tailed bat carcasses by sex found at Texas wind energy facilities.

Los Mirasoles (LM) (2017–2018, A–B), Los Vientos (LV) (2017–2018, C–D), and Wind Farm A (WFA) (2021–2022, E–F).

Discussion

California

The overall lack of bias in the sex ratio of Brazilian free-tailed bats at two wind energy facilities in California is consistent with what we know about this population’s presumed non-migratory behavior. The high proportion of male fatalities at one site (GHN) in June could be due to the local movement of male Brazilian free-tailed bats leaving roosting caves due to overcrowding as females move into caves and prepare for pupping in June (Davis, Herreid & Short, 1962). However, this pattern was not observed in June for GH, which is only 4 km from GHN, illustrated in Fig. 5. Under the assumption that the sex ratio of the non-migratory Brazilian free-tailed bat population in California is 50:50 (F:M), as seen in other non-migratory bat species (Davis, 1969; Cheng et al., 2019), it appeared that the proportion of female fatalities reflects the proportion of females in the population. Although our analysis does not specifically address this question, the absence of sex-specific bat-wind turbine behaviors in this non-migratory population could also explain these findings.

Figure 5 Locations of California wind energy facilities in study (Golden Hills and Golden Hills North).

Source Credits: California State Parks, Esri, HERE, Garmin, FAO, NOAA, USGS, EPA, SafeGraph, METI/NASA, Bureau of Land Management, NPS, USDA. Map Created by Sarah T. LiCari.

We considered two possible explanations for the shift in the timing of peak fatalities, which occurred in September and October at GH and May and June at GHN, as seen in both raw carcass counts and in the GenEst-corrected seasonal fatality monitoring estimates in the PCM reports for GH (H. T. Harvey & Associates (HTH), 2021) and GHN (Great Basin Bird Observatory (GBBO) & H. T. Harvey & Associates (HTH), 2022). One possibility proposed in the PCM report (Great Basin Bird Observatory (GBBO) & H. T. Harvey & Associates (HTH), 2022) is that GHN is near a reservoir used for foraging (Fig. 5), as Brazilian free-tailed bats are known to forage over large bodies of fresh lacustrine water during summer and fall (Johnston, 2013). Nevertheless, it was interesting that GH did not exhibit the same shift in 2017 and 2018, despite its close proximity to GHN (4 km apart), as depicted in Fig. 5, given that Brazilian free-tailed bats are known to fly over 56 km from their roosting sites when foraging at night (Best, 2009). Another possible factor that could have influenced this shift in timing is a change in moth abundance. Moths are a significant component of the Brazilian free-tailed bat’s diet (Kunz & Robson, 1995; Whitaker, Neefus & Kunz, 1996; Lee & McCracken, 2002), and bats have even been observed changing their foraging behavior in response to moth migrations (Krauel et al., 2018). Outbreaks of the California oak moth (Phryganidia californica) in May and June of 2019 and 2020 occurred in the same county in which the wind energy facilities in this study are located (Oboyski, 2020). The California oak moth experiences population outbreaks at irregular intervals that last one to two years (Burke & Herbert, 1920; Swiecki & Bernhardt, 2006). It is possible that GH also experienced a change in the timing of peak fatalities in 2019, but sample sizes were too small to detect a shift (n = 32). DNA barcoding of the stomach contents of the bat carcasses collected at GH and GHN could be used to determine if there were differences in their diets between locations (waterbody influence) or time periods (changes in moth abundance across years). Such data would help us better understand which external factors influence variation in bat fatalities at California’s wind energy facilities.

Texas

Estimates of the proportions of female Brazilian free-tailed bat fatalities varied across wind energy facilities and across time in Texas. The proportion of female fatalities differed between WFA and the two more southern locations, LM and LV, which are separated by only 2 km at the closest edge. Wind Farm A is approximately 200 km from LM and LV; it is possible that the sex ratio in the migratory Brazilian free-tailed bat population is not a uniform 9:1 (F:M; Davis, Herreid & Short, 1962) across the summer range. Differences in local sex ratios could have caused a higher proportion of male fatalities at WFA compared to the other wind energy facilities.

Previous studies regarding the Brazilian free-tailed bat sex ratio by roost type (bridge and cave) reported a difference in the proportion of females present in bridge roosts and cave roosts (Table S1). The average proportion of females at five surveyed bridge roosts was 0.44 (Sgro & Wilkins, 2003; Turmelle et al., 2010; Martinez, 2015). In contrast, the average proportion of females at 12 surveyed cave roosts was 0.73 (Cagle, 1950; Twente, 1956; Davis, Herreid & Short, 1962; Rogers, 1972; Thies, 1993; Turmelle et al., 2010; Danielson et al., 2022). Sex-specific roosting patterns may explain these differences (Hermanson & Wilkins, 1986; Sgro & Wilkins, 2003). For geographic reference, we have included known Brazilian free-tailed roost locations (bridges and caves), known bat roosts with assumed Brazilian free-tailed bats within the Texas range of the migratory population (S Weaver, 2023, personal communication), and roosts along the Mexico-United States border in the map provided in Fig. 6.

Figure 6 Locations of Texas wind energy facilities in study, known Brazilian free-tailed bat roosts, and known bat roosts with assumed Brazilian free-tailed bat populations by type (bridge or cave roost).

Data for known cave and bridge roosts were obtained from the literature. Data for assumed bridge roosts were provided by S. Weaver in 2023, personal communication. Assumed bridge roosts are roosts with known bat populations but have not been specifically studied for species occurrence. However, they are assumed to be Brazilian free-tailed roosts by the Texas Parks and Wildlife Department. Source Credits: Texas Parks & Wildlife, CONANP, Esri, HERE, Garmin, FAO, NOAA, USGS, EPA, NPS, Austin Community College. Map Created by Sarah T. LiCari.

Because of the sex bias in roost type utilization in this population, roost proximity could influence the proportion of female fatalities at WFA. Wind Farm A was adjacent to four bridge roosts, all within 50 km (Fig. 6), which could result in an estimated proportion of male fatalities that is higher than expected based on the overall population sex ratio. Additionally, WFA is 112.7 km north of a bridge roost location that males are known to use during the winter (Mink, 2012). Proximity to this roost could be causing the male skew we observed in September and October, as males travel from their bridge roosts north of WFA to this winter roost in southern Texas (Fig. 6). However, because these sites were surveyed in different years, we cannot exclude the potential impact of temporal variation in the population sex ratio on the observed differences in the proportion of female fatalities at the southern wind energy facilities (LM and LV) and the northern facility (WFA).

Temporal effects could also influence the differences in the sex ratio of fatalities across locations, as LM and LV were surveyed from 2017 to 2018, and WFA was surveyed from 2021 to 2022. We observed a change in the sex ratio over time at locations with multiple years of surveys, with LV showing a decrease in the proportion of female fatalities between the two years. Further research is required to ascertain if the decrease in the proportion of female fatalities is consistent across facilities or the result of inherent between-year variation and sampling limitations.

Conclusions

We hypothesized that the sex ratio of fatalities would be 0.5 in California, >0.5 in Texas, and that the peak timing of fatalities would occur between September and October in both states. Our findings confirmed that the sex ratio of fatalities in California did not differ from 0.5 as predicted, but the timing of peak fatalities shifted to earlier in the year (May and June) at GHN in 2019 and 2020. In Texas, the sex ratio of fatalities varied across different locations and over time. Although we did not have complete calendar year searches in Texas, for the time periods in which we did have carcass counts, the timing of peak fatalities appeared to be in September and October for all years and locations, as would be expected in this region (American Wind Wildlife Institute (AWWI), 2020). For this reason, a one-size-fits-all approach to curtailment is not suitable for either state due to inconsistencies in the proportion of female fatalities across locations and across time. For example, if all wind energy facilities in California were to implement a curtailment period during the typical peak season of fatalities in the fall, GHN would not have benefited from this in 2019–2020 due to the shift in the timing of peak fatalities to spring. Nonetheless, location-specific patterns in the proportion of female fatalities may vary over time, but further research is necessary to understand these patterns better. If implemented, continuous monitoring throughout the bat active season and across multiple years will help distinguish years of unusual activity for the species of interest that could be caused by external factors such as wildfires, droughts, or outbreaks of prey species. Additionally, standardization of PCM protocols and data collection could enhance current monitoring efforts and facilitate research on post-construction fatality monitoring studies across time and locations. Standardizing PCM protocols and data collection would also benefit wind energy facilities by reducing the cost of future monitoring and research (Conkling et al., 2021).

Supplemental Information

Supplemental Information 1 Raw Data.

Anonymized bat ID numbers, DNA Species ID, collection location (Country, State, County, and Wind Farm Name), Date Found, Estimated Time of Death, and Sex (Male: M, Female: F, and Unknown: X). The data was used to analyze the sex ratios of fatalities at wind energy facilities and compare variables.

Click here for additional data file.

Supplemental Information 2 Estimates of the proportion of female Brazilian free-tailed bats at bridge roosts and cave roosts from the literature using morphological sex identification.

All roosts fall within the range of the two US migratory populations, and cave counts were available for at least one month during the peak population abundance at these roosts in the United States (June to August). We calculated the average proportion of females for each roost type by averaging the proportion of female fatalities at each site within the category. We did not use the raw counts of male and female bats at roost sites to determine the average proportion of females across roost types to avoid introducing biases from sites with larger bat populations. We calculated significance using a one-proportion z-test (p0 = 0.5, α = 0.05). Symbols: * = statistically significant male-skewed, ** = statistically significant female-skewed.

Click here for additional data file.

We would like to thank the wind energy facility operators and consultants at the many California and Texas wind facilities for contributing samples to this study. Any use of trade, firm, or product names is for descriptive purposes only and does not imply endorsement by the U.S. Government.

Additional Information and Declarations

Competing Interests

Author Contributions

Field Study Permissions

Data Availability

David M. Nelson is an Academic Editor for PeerJ. Amanda M. Hale is employed by Western EcoSystems Technology, Inc. Sara P. Weaver is employed by Bowman Consulting Group.

Sarah T. LiCari conceived and designed the experiments, performed the experiments, analyzed the data, prepared figures and/or tables, authored or reviewed drafts of the article, and approved the final draft.

Amanda M. Hale conceived and designed the experiments, authored or reviewed drafts of the article, and approved the final draft.

Sara P. Weaver conceived and designed the experiments, authored or reviewed drafts of the article, and approved the final draft.

Sarah Fritts conceived and designed the experiments, authored or reviewed drafts of the article, and approved the final draft.

Todd Katzner conceived and designed the experiments, authored or reviewed drafts of the article, and approved the final draft.

David M. Nelson conceived and designed the experiments, authored or reviewed drafts of the article, and approved the final draft.

Dean A. Williams conceived and designed the experiments, analyzed the data, authored or reviewed drafts of the article, and approved the final draft.

The following information was supplied relating to field study approvals (i.e., approving body and any reference numbers):

The samples in this study were provided to us by the Renewable Energy Wildlife Research Fund (REWRF). The samples used in the project originated from Post Construction Fatality Monitoring surveys at wind energy facilities in California and Texas. In California, samples were collected by H.T. Harvey & Associates (HTH) under a permit from the California Department of Fish and Wildlife (SC-4607). The samples in Texas were collected by Texas State University or Bowman Consulting Group in accordance with Texas Parks & Wildlife Department (TPWD) permit numbers SPR-1120-189 and SPR-0213-023. The samples were then sent to Texas Christian University for analysis.

The following information was supplied regarding data availability:

The raw sex identification data of bat fatalities at wind energy facilities is available in the Supplemental File.

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
