# Peer review of "Understanding fatality patterns and sex ratios of Brazilian free-tailed bats (Tadarida brasiliensis) at wind energy facilities in western California and Texas"

_PeerJ, doi:10.7717/peerj.16580_

## Round 0.1 · original submission · Major Revisions

Please incorporate all comments of reviewers and submit the revised version along with a point-to-point rebuttal letter. Please improve the literature section by citing very recent papers in the subject area. Moreover, please improve the language of the paper.

Reviewer 1 ·

Basic reporting

The manuscript is well written, and appropriate citations are used properly throughout.

Experimental design

I wish that sampling was done more consistently across the sites over time, but I understand that is not always possible. Still, the methods seem valid and analysis reasonable. Assumptions made in the case of incomplete coverage or other statistical limitations were also reasonable.

Validity of the findings

So little is known about seasonal Tadarida brasiliensis movements across the landscape that this study represents an important addition to the literature.

The maps showing known roosts in Texas are helpful in interpretation of the results. If the authors could possibly include approximate location of known roosts of any kind over the Mexican border within foraging or short-range landscape movement, that would be even more helpful. During the autumn period bats may be moving around the landscape prior to migration, and if migration is sex biased this could have an effect on interpretation. On the other hand, almost nothing is known about how these bats move seasonally across the landscape in California and these results are very interesting.

Additional comments

In Fig 2, it might be easier to follow if you switch C and D (2019 and 2018) so that years increase left to right.

·

Basic reporting

This study uses Tadarida brasiliensis carcass data from wind energy facilities in Texas and California to infer the sex ratios of these bats impacted at these facilities, and evaluates a priori hypotheses about migration behavior and sex ratios in these areas. The manuscript is well written with a short introduction to the problem and the relevant natural history characteristics of these bats.

Experimental design

The study design is appropriate for the questions addressed in this manuscript.

Validity of the findings

The findings appear to be valid, based on the description of the methods and results.

Additional comments

The following are my comments on the manuscript. Thanks for letting me take an early look at this manuscript. I enjoyed reading it.

The carcass data collected and sex determination methods used seem to be of high quality and appropriate for evaluating sex status of carcasses. I agree with the authors that they do not need to derive formal abundance estimates, such as would be derived using GenEst, etc.

I would encourage the authors to consider finding and using some more recent citations to support their claims about migratory behavior for this species ad related topics. In several spots in the manuscript the authors rely on citations from the 1950’s and 1960’s to describe migratory and colony-forming behavior (e.g., line 113 and similar). There are more recent citations that could be used to support these claims, such as Ammerman et al's Bats of Texas (2012), etc. regarding this species in Texas. There are other more recent peer-reviewed papers that can be cited for the migratory behavior and patterns exhibited by this species. I would encourage the authors to review the manuscript with an eye toward using the most recent high quality citations available. I like that the authors are clearly familiar with the historical literature on this species, but would encourage them to also include more up-to-date citations. When only citations from the 1950’s and 1960’s are used, it leaves readers with the impression that there hasn’t been any meaningful work done in that particular area since that time, which is not the case. In fact, there is a quite substantial more recent literature on the migratory patterns of this species.

I would also encourage the authors to consider addressing the following questions that came to mind for me as I read the manuscript:

- Line 71, “bat collisions with wind turbines”. Consider using the phrase “bat fatalities” here. It is sometimes not clear if fatalities are caused by collisions or barotrauma, etc.
- Line 72. I believe the American Society of Mammalogists has recently suggested changing the genus names of some of these migratory tree bat species. Check to confirm that the genus names used represent current nomenclature and consider using both genera names if there is any confusion.
- Line 80, “species of concern”. This phrase is sometimes used more formally in some states to designate a species that is of conservation concern, or threatened or endangered. Consider rephrasing to eliminate possible confusion, such as “Some biologists have expressed concern…” or something similar.
- Line 80. Consider adding more recent citations here.
- Line 95 pesticide use citations. Consider adding more recent citations.
- Line 100, “smart curtailment strategies”. Consider adding a citation or two here.
- Line 111. Consider adding more recent citations.
- Line 115-117. Migration is not the only behavior that could influence sex ratios at a wind facility. For example male-skewed seeking of large objects on the landscape to set up possible mating territories, etc., could influence these patterns. Consider adding some detail to this paragraph with some possible alternative explanations, along with appropriate citations.
- Line 123. Consider adding more recent citations.
- Line 131. Consider adding more recent citations.
- Line 151. For this and other wind turbine descriptions consider adding the height, blade length and model of the turbine.
- Line 153. Consider adding a citation related to the climate description for interested readers.
- Line 155. September 2015 through September 2017 is listed as 1 year.
- Line 158. GRTS description. Consider adding enough detail on how you used GRTS so a reader could reproduce your methods. As is, I’m not sure I could reproduce what you did with your GRTS sampling. If necessary you could consider adding some detail in supplementary materials.
- Line 161. Consider adding a line or two about how dog teams searched plots, and describe if you evaluated searcher efficiency among dogs. My understanding is that dog teams can exhibit substantial differences in searcher efficiency. Consider describing how this was addressed in this project.
- Line 163. Consider stating the size of wing punches used for wing samples. I’m assuming they are wing punches.
- Line 262. “no significant difference”. Consider adding the name of the statistical test conducted here along with the associated statistics, if available.
- Line 293. Consider adding more recent citations.
- Line 300. I am not following why 30 bats was used here. Consider clarifying.
- Line 303. Consider stating when the Bonferroni correction was applicable. Some readers will not be clear on this.
- Line 326. Consider using “presumed non-migratory…”
- Line 357. Consider clarifying “migratory behavior” as “presumed non-migratory…” or something similar.
- Line 433. Consider adding an example of what a “one-size-fits-all” curtailment approach would look like.
- Table 1 seems like it could be used in supplementary materials and may not be necessary in the main paper.

Reviewer 3 ·

Basic reporting

The manuscript is clearly written and composed with a professional tone. Stylistically, I have no substantial comments. Though there are many different aspects to the methods, the authors have done a good job explaining these differences and their choices. I found the introduction to provide sufficient background information and to justify the need for the study. The structure of the manuscript is typical and appropriate however seven figures plus a table fells a bit excessive to me. I would recommend omitting table 1 as molecular sex determination is quite routine. Table one is useful to a degree, but perhaps would be better as an appendix (if its purpose is to substantiate that bridge roosts are I believe this is accomplished by the citations in the discussion). The raw data are shared and I would seem to allow for independent verification of the results presented within. The manuscript's focus is narrow (a good thing) and it sticks to that focus throughout (table 1 being an exception, in my opinion).

Experimental design

The objectives of the research within the manuscript are within the Aims and Scope of the journal. The research questions poses (sex-biased mortality at wind energy facilities is well defined, relevant, and meaningful. All mortalities are nor equal, especially when a species' life history included large aggregations of one sex. Results form this research would help fill an existing knowledge gap about the sex-biased impacts of wind energy facilities on bats (or at least this species).

I think it is important to acknowledge the difficult logistic nature of of this investigation and to commend the authors for their efforts. Many of the differences in the post construction monitoring between CA and TX were likely out of their hands and i believe they have done a good job explaining how sampling occurred between the two areas. Based on my reading, it is unlikely that the differences in surveys for and recovery of bat carcasses influenced the results. Further, the authors have tried to leverage carcasses from wind turbines and extract additional value (sex ratios) from these carcasses.

DNA extraction and molecular sex determination are in line with other studies.

Within the data analysis it is my interpretation that the procedure described in lines 258-270 (retaining samples even when ETD data was missing) only refers to the analyses on TX data. Between lines 261 and 262, the last part of line 261 feels out of place - should it be the beginning of the sentence at line 262? I'm not sure what to make of this. It appears as it the authors preferred using the more restrictive data set (with ETD data) but, due to the number of records without the ETD data (n = 187) decided to use the full TX dataset. Given that there is no ETD data for any of the CA records and, given that there are enough differences between the CA surveys and the TX surveys, why try to introduce another one (we used ETD in TX but date found in CA). It just seems a complicating factor. I would suggest removing the first paragraph under data analysis all together.

Validity of the findings

The findings from CA are straightforward and are supported by the data. Explanations for the unexpected peak fatalities between May and June (2019 and 2020) are entertained and the scenario of the California oak moth superabundance seems quite plausible.

I think that if the brief discussion about ETD data is removed from lines 258-270, the interpretation and validity of the TX data are quite simple. The way that paragraph is written, an uncritical reader might accuse the authors of cherry picking their data for analyses. This would be wrong in my view as the authors did not restrict what data was analyzed (censoring unwelcomed data points) but rather expanded the data used in the analysis and thereby had more power to detect differences in sex ratio. What then remains is a fairly strong bias (towards males) at the WFA location and then no or vert weak bias toward females at LM and LV, respectively. The location of male-biased roosts (bridges) near the WFA location lends some support to this as do the absence of female-biased roosts (caves) near LM and LV.

Moreover, I think it appropriate that your discussion focuses more on the male-bias at WFA rather than the weak female-bias at LV. Though there is a statistically significant difference at LV, its rather small and could be a temporal anomaly or caused by some unknown confounding factor.

Additional comments

I enjoyed reading your manuscript. Despite the daunting task of bringing together two different carcass collections methods across two different states, I think you manuscript reads well. Aside from the comments above, I've made a few suggestions to the text below.


Abstract
- Line 28: would suggest 'informed curtailment' over 'smart curtailment'.
Methods, Data Analysis
- Line 260: delete 'had'
Results
- line 319: these number do not add up. When I summed them i received 1737. The text on lines 320 and 321 (90 bats unbaled to be assigned sex so a total of 1,647 in analysis) also suggests the correct number to be 1,737.
- Line 329; suggest a space between the colon and parenthesis
Discussion
- Line 359: suggest commas to between each of you analysis groups (e.g., …analyzed by year, by location and month, and by location...

---

## Round 0.2 · accepted · Accept

The paper has been improved incorporating all comments of reviewers and is accepted for publication.

·

Basic reporting

The reporting in this revision is well done.

Experimental design

The study design is appropriate for the questions addressed in this manuscript.

Validity of the findings

The findings appear to be valid, based on the description of the methods and results.

Additional comments

The authors have done a great job of addressing my original comments.